# OpenReview forum: "The Negation Bias in Large Language Models: Investigating bias reflected in linguistic markers"
_colmweb.org/COLM/2025/Conference — COLM 2025_

### Official Review · Reviewer_qKBW · 2025-05-01

**Rating:** 7
**Confidence:** 4
**Ethics Flag:** 1

**Summary:**

In this work the authors present an experiment to investigate whether LLMs exhibit negation bias, that is, a bias towards negating a positive trait as opposed to using a more negative term directly.

Overall, I think this is a very nice paper that does a really good job on a very specific experiment. Beyond the specific experiment, I think the authors have an opportunity to zoom out and present this work as a type of template that others could follow when trying to probe LLMs in similar ways.

**Questions To Authors:**

* Can you add a figure to lay out the process? I think that would really help with my 2nd point in the summary and also to just make sure that the reader, who may not be as familiar with social science type experiments, is clearly guided through the process.
* Interaction is significant for decoder models but not for encoder models. Can you elaborate on this finding? Or use larger MLMs?
* It would be nice to link this work to prior work on probing in machine learning (e.g., [1-3])

References

1. Ritter, S., Barrett, D. G., Santoro, A., & Botvinick, M. M. (2017, July). Cognitive psychology for deep neural networks: A shape bias case study. In International conference on machine learning (pp. 2940-2949). PMLR.
2. Lalor, J. P., Wu, H., Munkhdalai, T., & Yu, H. (2018, October). Understanding deep learning performance through an examination of test set difficulty: A psychometric case study. EMNLP (Vol. 2018, p. 4711).
3. Dasgupta, I., Lampinen, A. K., Chan, S. C., Sheahan, H. R., Creswell, A., Kumaran, D., ... & Hill, F. (2022). Language models show human-like content effects on reasoning tasks. PNAS Nexus, Volume 3, Issue 7, July 2024, pgae233, https://doi.org/10.1093/pnasnexus/pgae233

**Reasons To Accept:**

* Line 231 explicitly defines the hypothesis being tested, that’s great!
* A carefully designed analysis of a specific phenomenon with a link to social science research.
* This experimental setup could be applied to other forms of bias. At the moment I don’t think this is emphasized enough though (see below).

**Reasons To Reject:**

* Using larger MLMs would make for a more appropriate comparison
* (very minor) It’d be nice to see how negation bias fits into a larger framework of biases, so that we can tell how/if these results are generalizable/can be adapted to address other forms of bias.

---

> ### Author Response · Authors · 2025-06-02
>
> Thanks for your review. We agree larger MLMs and/or smaller decoder models would make a more appropriate comparison, as also suggested by reviewer iwy7. To address this concern, we ran some additional experiments using SmolLM models for smaller decoder models and EuroBERT as larger encoder models:
>
> * For larger MLMs, we experimented with the recent EuroBERT, which also goes up to 2.1B parameters, although it is multilingual. These models do not exhibit the bias. EuroBERT-2.1B (coef = -0.199, p=0.548), EuroBERT-610M (coef = -0.471, p=0.080) and EuroBERT-210M (coef = 0.378, p=0.245) do not show significant interactions between the variables of form and stereotypicality. Despite testing this recent model in different sizes, there is no clear effect of size.
> * We find the effect for SmolLM 135M (coef = 7.832, p=0.004) and 350M (coef = 5.742, p = 0.001). For 1.7B, we do not find a statistically significant effect, although it is close (coef = 2.431, p = 0.08). This suggests that model size is not necessarily the main factor for decoder models.
> * We also experimented with DeBERTa-v2 large models which are large and more monolingual, however, we found that we get very high pseudo-perplexity values overall with these models (over 100000, rather than the usual 1-100 range). Perhaps pseudo-perplexity-per-word is not a reliable metric for these models or our implementation of it doesn’t work for DeBERTa, so we still have to validate the metric for these models. With these high perplexity values, we do not find a significant result for deberta-v2-xlarge (coef = -2129.02, p=0.754), but we do find a significant result for deberta-v2-xxlarge (coef = 8924.18, p=0.002). This would then be the only encoder model exhibiting the bias, if the result is valid, which we still have to confirm with error analysis.
>
> If accepted, we will include these results (in more detail) to strengthen the evidence for our findings, and include a discussion of relevant literature on model size and bias.
>
> We agree that it is important to position this bias better in the larger typology of biases, as other reviewers were also not quite clear on what kind of bias this is. If accepted, we will contextualize it with regards to other linguistic biases such as the linguistic intergroup bias, the stereotypic explanatory bias, specificity bias, use of nouns vs adjectives when describing persons, and the connection of these biases to stereotypicality. This will also allow us to explain how the approach can be applied to other linguistic biases.
>
> We do have space for an extra plot for positioning negation bias within a wider context of social biases. We have designed a summary figure, which we are not able to attach to this comment, and will add it to the paper if accepted. The image shows Negation Bias, Modality Bias and Linguistic Intergroup Bias with for each of these biases example context sentences, example pair sentences and the linguistic marker of the bias.
>
> Linking this work to prior research on probing in machine learning: This is challenging because this is a very broad area. There are many probing methods which are often specific to tasks or model architectures. However, we agree it’s a valuable perspective and will revisit how to position our approach within the broader literature on probing methods.To clarify the general approach we can certainly position our perplexity-based approach among other related probing approaches, which include targeted masked language modelling probing, probing experiments inspired by behavioural experiments as well as mechanistic interpretability experiments. If we have space left we will do this.

---

> > ### Comment · Reviewer_qKBW · 2025-06-05
> >
> > Thank you for the reply. I look forward to seeing the updates you described.

---

### Official Review · Reviewer_iwy7 · 2025-05-13

**Rating:** 7
**Confidence:** 4
**Ethics Flag:** 1

**Summary:**

Authors introduce a new framework for examining stereotypical biases encoded in LLMs: claiming that the majority of existing bias research focuses on surface-level associations, they use a well-known "negation bias" in humans to examine if this bias exists similarly in LLMs. Authors bring concepts from social science bias research into the NLP/ML domain. They test on both masked language models (MLMs) and large autoregressive models. They find that the autoregressive models show evidence of this same bias humans have, but the MLMs do not. The paper is well-written and clear.

**Reasons To Accept:**

Authors introduce a new bias evaluation framework that has, to my knowledge, not been examined in the literature directly (looking at the negation bias as an indirect marker of sterotypical bias). This avoids the pitfalls of the majority of existing bias literature, which looking at associations at the surface-level for the most part.

Authors expand/modify an existing dataset from another paper (Beukeboom et al 2020) to construct a new negation bias dataset to investigate whether the LLMs exhibit the same bias as humans.

Authors integrate methods across disciplines (social sciences and ML/NLP) in an interesting way.

**Reasons To Reject:**

The paper does not do any investigation to disentangle various factors involved in WHY the MLMs don't exhibit the bias while the autoregressive LLMs do. Specifically, they acknowledge the confounding variable of model size. I think this huge confounder is a significant problem in the paper. I believe it would have been relatively straightforward for the authors to "standardize" model size and remove it as a variable. Models like SmolLM (135M, 360M, and 1.7B parameters) exist for the "smaller autoregressive LLM" category, and large MLMs also exist (DeBERTa V2 xxlarge is at 1.5B parameters). This would have been a very interesting comparison to make.

The conclusion of the paper feels relatively predictable. The finding that larger models more accurately reflect statistical patterns in the training data (which also includes biases) feels like a somewhat "known"/predictable finding, as they have the capacity to encode more subtle patterns. Without disentangling model size from the experimentation, it is hard to see why it isn't reasonable to conclude this difference that the authors observed is mostly or even entirely due to this factor. To be even more specific, here are a couple of existing works that tie model size to bias:

https://arxiv.org/abs/2206.09860

https://aclanthology.org/2023.emnlp-main.161/

and I will posit that this is a somewhat "well-known" effect generally. This thesis deals with it directly: https://www.ideals.illinois.edu/items/131343

The claim about "expanding bias evaluation beyond explicit associations" needs a lot of nuance. There are significant works in bias/safety in NLP that already do this, for example the CrowS-Pairs dataset that authors cite in the paper itself. The paper specifically examines expanding bias evaluation beyond explicit associations *via analyzing indirect linguistic markers*. This is interesting, but I think the paper could discuss the existing works more that are beyond explicit associations and contextualize their work more effectively. it feels like the authors are painting a somewhat "out of date" picture of the rest of the literature as it currently stands.

---

> ### Author Response · Authors · 2025-06-02
>
> Thank you for your review, especially the constructive advice. The current set of models was selected to cover a range of architectures, sizes, and families from widely used models. However, we agree that further size-controlled models would strengthen the analysis. Thanks for the model suggestions! We have experimented with these models as well, and found the following:
>
> * We find the effect for SmolLM 135M (coef = 7.832, p=0.004) and 350M (coef = 5.742, p = 0.001). For 1.7B, we do not find a statistically significant effect, although it is close (coef = 2.431, p = 0.08). This suggests that model size is not necessarily the main factor for decoder models. We will include these results in more detail in the paper, if accepted.
> * We also experimented with DeBERTa-v2 large models, however, we found that we get very high pseudo-perplexity values overall with these models (over 100000, rather than the usual 1-100 range). Perhaps pseudo-perplexity-per-word is not a reliable metric for these models or our implementation of it doesn’t work for DeBERTa, so we still have to validate the metric for these models. With these high perplexity values, we do not find a significant result for deberta-v2-xlarge (coef = -2129.02, p=0.754), but we do find a significant result for deberta-v2-xxlarge (coef = 8924.18, p=0.002). This would then be the only encoder model exhibiting the bias, if the result is valid, which we still have to confirm with error analysis.
> * As an alternative, we experimented with EuroBERT, which also goes up to 2.1B parameters, although it is multilingual. These models do not exhibit the bias. EuroBERT-2.1B (coef = -0.199, p=0.548), EuroBERT-610M (coef = -0.471, p=0.080) and EuroBERT-210M (coef = 0.378, p=0.245) do not show significant interactions between the variables of form and stereotypicality. Despite testing this recent model in different sizes, there is no clear effect of size.
>
> If accepted, we will include these results (in more detail) to strengthen the evidence for our findings, and include a discussion of relevant literature on model size and bias.
>
> If you have any other suggestions for factors that need to be disentangled beyond model size, we’d be very happy to hear them!
>
> Regarding the prior bias literature you mentioned, we want to emphasize that the novelty of our work lies in examining subtle, structural biases expressed through linguistic form — for example, a higher use of negation when a statement goes against a stereotype. Another example of this type of subtle bias is how people choose between abstract and concrete language when describing others. Research on linguistic intergroup bias (LIB) shows that people tend to describe positive behaviors by in-group members and negative behaviors by out-group members in more abstract terms (e.g., “She is kind,” “He is aggressive”), while describing behaviors that contradict these expectations in more concrete terms (e.g., “She didn’t help her friend”) (Maass et al., 1995) If accepted, we will contextualize the negation bias with regards to other linguistic biases such as the linguistic intergroup bias, the stereotypic explanatory bias, specificity bias, modality bias, use of nouns vs adjectives when describing persons, and the connection of these biases to stereotypicality.
>
> Studying these kinds of biases requires carefully controlled experimental materials that isolate specific linguistic features. However, we find that much of the current literature lacks this level of control. For instance, while datasets like CrowS-Pairs are valuable, they tend to focus on surface-level word associations and use crowdsourced sentences about social groups, without systematically targeting or analyzing deeper linguistic structures like negation or abstraction. CrowS-Pairs directly targets social bias categories, and we are not aware of any categories or controls of indirect biases expressed through linguistic form. As far as we are aware, the pairs only show different values for social bias categories, rather than including pairs with different linguistic forms as we do. If you have the time, could you please show us an example from CrowS-Pairs including indirect linguistic markers that illustrates what you mean? Or if you have any other references to work that does this, we would be happy to receive them. We are not aware of any such datasets.
>
> Either way,  we will clarify this distinction more clearly and better situate our contribution within the broader bias literature.
>
> Maass, A., Milesi, A., Zabbini, S., & Stahlberg, D. (1995). Linguistic intergroup bias: differential expectancies or in-group protection?. Journal of personality and social psychology, 68(1), 116–126. https://doi.org/10.1037//0022-3514.68.1.116

---

> > ### Comment · Reviewer_iwy7 · 2025-06-07
> >
> > I thank the authors for the additional experiments and clarifications. I look forward to the additional experiments being added to the paper, and am raising my score.

---

### Official Review · Reviewer_Ykig · 2025-05-13

**Rating:** 5
**Confidence:** 3
**Ethics Flag:** 1

**Summary:**

This research paper studies negation bias in LLMs. Negation bias is when people use negative statements (like "not tidy") instead of direct descriptions (like "messy") when describing situations that go against stereotypes. The paper created special sentences with stereotypical and non-stereotypical contexts, plus affirmed and negated descriptions. The paper tested 8 different language models by measuring "perplexity" (how surprised the model is by certain text). Results show that autoregressive models (GPT, Llama-3.1, Mistral) show the same negation bias humans have, while encoder models do not show this bias. This suggest LLMs might perpetuate subtle biases through language patterns, not just through direct stereotypes. The paper claims that we need more testing for these subtle linguistic markers of bias in language models.

**Questions To Authors:**

### Suggestions
* Figure 1: "h" in "housewife" is not boldened.
* Figure 2: The plot is too small. This is problematic especially given that COLM allows 9 pages maximum for the main text and 1 full page can be used to expand this plot.
* Lines 58: "eight state-of-the-art models" sounds odd to me since it's claiming both BERT-base and Llama 8B as SOTA. Suggest rephrasing it.
* Limitations Section: Experimented model size (<=8B) is an important limitation of this work. Please also be explicit about it.

### Questions
* Figure 2: The error bars are relatively large and showing the results on stereo vs. nonstereo setups are within the error bar. Isn't this indicator of not much difference between the setups and no conclusive takeaway can be claimed from these plots?
* Appendix A: E.g., Are "truely honest" vs. "not sneaky" semantically the same? If this is "honest" vs. "not honest" or "sneaky" vs. "not sneaky", I would be more convinced with it but seems like there's also a semantic shift here in addition to linguistic form change.
* Why are masked language models "show no significant interaction between context and form" (Line 261)?

**Reasons To Accept:**

* Analysis of LLM biases from negation bias perspective.
* A new dataset for evaluating negation bias in LLMs.

**Reasons To Reject:**

* Experimented model size (<=8B) and variants (no instruction tuned models) are limited. Also proprietary models are not experimented.
* There's a questionable setup in linguistic form (Affirmation vs. Negation) difference.

---

> ### Author Response · Authors · 2025-06-02
>
> Thank you for your thoughtful feedback. Our goal is to provide a framework for evaluating subtle bias in LMs, using negation bias as a starting point. The selected models serve as illustrative examples and not an overview of the state of the art. If accepted, we can include Llama 3.3 70B (instruct and non instruct) as a representative example of a large decoder model. We will also acknowledge the limitations in model size (≤8B) for most models and the lack of proprietary models and lack of comprehensive overview of all models. We also added EuroBERT and SmolLM models based on other reviewer comments (please see other comments for results using these models).
>
> We appreciated the detailed suggestions, which are very valuable. We'll fix the formatting in Figure 1 and enlarge the text in Figure 2, and revise the phrasing of the sota model and clarify the Limitations section accordingly.
>
> “There's a questionable setup in linguistic form (Affirmation vs. Negation) difference.” - Could you please clarify this point? It is not clear to us what is questionable about our setup. If you could express more specifically what aspect of it is questionable, that would be very helpful for our revisions.
>
> On your questions:
> * Figure 2: We rely on inferential statistical testing with linear models to establish significance of results. While error bars are large, there are significant interaction effects in three models. Further, some error bars does not overlap (eg. Mistral 7B, non-stereotype). We cannot read from error bars alone whether an effect is significant. These are confidence interval (CI) error bars, and CI error bars do not consider standard error per category, only individual standard error, among other reasons (Lanzante, 2005). This is why we perform inferential statistical testing (with the coefficients and P-values) to establish whether our results are indicative of differences between the setups.
> * Semantic shifts: By semantically the same we indicate the phrase expresses the same meaning. ‘Good’ would have a similar meaning as ‘not bad’. We can indeed question the extent to which paired expressions are truly the same, as per the Principle of No Synonymy (Goldberg, 2006) there are always subtle differences between linguistic expressions. Either way, these test items are taken from Beukeboom et al. (2020), and in that study, native speakers validated the test items. We chose not to modify them in order to be consistent with other work on the negation bias. We will make this information explicit in the paper, if accepted.
> * Line 261: We draw the conclusion from our linear regression analysis (Table 1). We will make the claim more clear by saying ‘Our analysis suggests that, for the BERT-based models (BERT-Base, BERT-Large, ModernBERT, and ModernBERT-Large), there is no significant interaction between context and form in predicting perplexity. This indicates that negation does not differentially affect perplexity in different contexts, providing no evidence of a negation bias in these models.’
>
> Goldberg, A. E. (2006). Constructions at work: The nature of generalization in language. Oxford University Press.
>
> Lanzante, J. R. (2005). A cautionary note on the use of error bars. Journal of Climate, 18(17), 3699-3703.

---

> > ### Comment · Reviewer_Ykig · 2025-06-10
> >
> > I would like to thank the authors for detailed response and clarification.
> >
> > > “There's a questionable setup in linguistic form (Affirmation vs. Negation) difference.” - Could you please clarify this point? It is not clear to us what is questionable about our setup. If you could express more specifically what aspect of it is questionable, that would be very helpful for our revisions.
> >
> > Thanks for the clarification and let me clarify. This is the same point as the second question on "Appendix A: E.g., Are "truely honest" vs. "not sneaky" semantically the same?". And just to clarify, as long as these assumptions (and where it originated from, in this case Beukeboom+ 2020) are explicit in the manuscript, I believe it's fine.
> >
> > > We cannot read from error bars alone whether an effect is significant.
> > I agree with that error bars alone does not show statistical significance or not, but also be cautious about overly relying on p-values (which itself can be easily hacked) given that the dataset size is relatively small i.e., 300 examples (Line 150). Isn't the large error bars indicating that the sample size is small?
> >
> > > We will make the claim more clear by saying ‘Our analysis suggests that, for the BERT-based models (BERT-Base, BERT-Large, ModernBERT, and ModernBERT-Large), there is no significant interaction between context and form in predicting perplexity. This indicates that negation does not differentially affect perplexity in different contexts, providing no evidence of a negation bias in these models.’
> >
> > "no evidence of negation bias" based on non-statistical significance (i.e., the evidence is not strong enough to reject null hypothesis) in interaction is overclaiming and misinterpreting the results. I would suggest removing the last sentence.

---

> > > ### Author Response · Authors · 2025-06-10
> > >
> > > Thanks for your follow-up reply! In revisions, we will be clear about where the assumptions built into the dataset come from.
> > >
> > > Error bars: Well, it is an interaction between sample size and effect size. The smaller the effect, the larger the sample size needed to prove it. In theory we could make this more explicit by including a power analysis, but there is no reliable way to do power analysis in a mixed effects model apart from doing simulations. Either way, by also reporting the effect size we allow the reader to interpret the practical significance of the effect.
> > >
> > > Claims about non-significance: You are right that it is very important not to overclaim. One cannot prove a null hypothesis, however, by stating that we find "no evidence of negation bias", we think we aren't doing that. If we said "our results show there is no negation bias" this would be bad, as you cannot prove a null hypothesis (e.g. "there is no negation bias"). However, we say "no evidence of negation bias" meaning that our experiment does not find evidence (but an experiment with a larger sample size could show a different result). So, we believe that we are doing correct statistical reporting by phrasing it in this way, and this is how I've always learned to draw conclusions about null results (while knowing that many people and many published papers do this incorrectly). But if you have a more accurate suggestion, we'd be happy to hear it! We can be more specific about the possibility that a larger experiment could find the bias.
> > >
> > > As the discussion period is coming to an end, if our comments have changed your mind about aspects of our work, we'd appreciate if you would consider updating your score to reflect the discussion.

---

### Official Review · Reviewer_gxew · 2025-05-13

**Rating:** 7
**Confidence:** 4
**Ethics Flag:** 1

**Summary:**

They use a multi-disciplinary approach, borrowing ideas from social science, to study social biases in LLMs.

They found autoregressive decoder models to have negation bias which is not present in encoder models like BERT.

Their findings shows that current bias evaluations may underrepresent the social biases in models by overlooking linguistic markers such as negation.

**Questions To Authors:**

- I think this is a solid work and I would love to see what authors think on how this can be expanded to test more subtle biases (broadening scope beyond negation bias) in LMs.

**Reasons To Accept:**

- These kind of multi-disciplinary studies are very much needed in the community to better understand how subtly social biases are rooted in LMs
- Related Works section is well structured and gives a good overview of the field
- Paid attention to detail by taking into account length dependency for comparison using perplexity scores. Fixed the issue by adding "truly" word in affirmative sentences.
- The data aug of using Synonym Replacement and Sentence Structure Variation for dataset creation shows that authors great attention to detail.

**Reasons To Reject:**

- I think using pseudo-perplexity for MLMs and comparing them with perplexity scores with LMs is not fair as I expect perplexity scores to be consistently higher as compared to pseudo-perplexity as MLMs have more/bi-directional context => higher probs => lower pseudo perplexity. This might be the reason why authors such varied results for LMs vs MLMs,

---

> ### Author Response · Authors · 2025-06-02
>
> Thank you for your review. We appreciate your recognition of the importance of studying subtle social biases in language models and your positive feedback on the structure and methodological attention in our work.
>
> Regarding your concern about the comparison between perplexity scores for causal LMs and pseudo-perplexity for masked LMs: we agree that these two metrics are not directly comparable due to the differing architectures and context usage of the models (e.g., unidirectional vs. bidirectional context). That’s why we do not compare perplexity values from different models directly anywhere in our study: we only compare conditions within models, and then we only compare the relative differences between conditions through coefficients in linear models. The absolute scaling of perplexity or pseudo-perplexity values does not matter when we compare coefficients of those values in a linear model, as far as we can tell, and even if they did, we do not compare them directly between models. Furthermore, our decision to include both perplexity and pseudo-perplexity was informed by precedent in prior work, where these metrics are also used to compare across conditions in the same model. We will state more explicitly in the paper that there is no direct comparison of the two types of values. Or if we misunderstood your concern, please let us know and we can address the follow-up comment.
>
> On the question of extending this research beyond negation bias: we completely agree that there is rich potential to explore more subtle biases and other linguistic biases. Negation bias was chosen in this initial work because there is a well-constructed dataset in the communication science literature. As you noted, linguistic abstraction (as discussed in Beukeboom & Burgers, 2019) is another compelling direction—particularly because abstraction levels can encode stereotypes in implicit ways. Other similar biases established in the sociolinguistics literature are the linguistic intergroup bias, the stereotypic explanatory bias, specificity bias, modality bias, and the use of nouns vs adjectives when describing persons. Since other reviewers raise this as well, if accepted, we will contextualize negation bias by also describing related linguistic biases and their connection to stereotypical bias.

---

> > ### Comment · Reviewer_gxew · 2025-06-07
> >
> > I thank the reviewer for addressing the concerns. After reading the comments, I would like to maintain my score.

---

### Decision · Program_Chairs · 2025-07-08

**Decision:**

Accept

**Comment:**

This paper creates a new bias evaluation benchmark for stereotypes (+analysis and results on current encoder models and LLMs) based on the presence of negation which has been attested as a marker of stereotypical vs. antistereotypical statements in human speech. The paper is statistically rigorous (refreshing in the ML world) and grounded in psycholinguistic theory.
However, the takeaways, though valuable, are not incredibly surprising (that larger models pick up more patterns from corpora) and the causal nature of this isn't addressed.

The paper does have extensive related work, but also neglects to reference or situate the work within common and known pitfalls of templated solutions: e.g. https://arxiv.org/abs/2210.04337 on the brittleness of these methods, which is even more attested in the sensitivity of LLMs to prompts.
More surprisingly, the paper does cite Blodgett 2021 https://aclanthology.org/2021.acl-long.81/ but doesn't engage with the critique of stereotype datasets. As is what to take away from the results -- the paper sits oddly between a computational social science paper and an LLM paper.

So it overall is a well down paper, but the lack of engagement with methodological pitfalls of this approach found in previous work limits the impact it can have on the field. This bothers me more than the reviewers though, who broadly approve of the paper (though they perhaps are unaware of these limitations as they are not mentioned).
If the paper is accepted, I recommend at least addressing this in writing, though it would be preferable to address with ablations.